# Deep Transformation-Invariant Clustering

**Tom Monnier**        **Thibault Groueix**        **Mathieu Aubry**

LIGM, Ecole des Ponts, Univ Gustave Eiffel, CNRS, France
{tom.monnier,thibault.groueix,mathieu.aubry}@enpc.fr

## Abstract

Recent advances in image clustering typically focus on learning better deep representations. In contrast, we present an orthogonal approach that does not rely on abstract features but instead learns to predict transformations and performs clustering directly in pixel space. This learning process naturally fits in the gradient-based training of K-means and Gaussian mixture model, without requiring any additional loss or hyper-parameters. It leads us to two new deep transformation-invariant clustering frameworks, which jointly learn prototypes and transformations. More specifically, we use deep learning modules that enable us to resolve invariance to spatial, color and morphological transformations. Our approach is conceptually simple and comes with several advantages, including the possibility to easily adapt the desired invariance to the task and a strong interpretability of both cluster centers and assignments to clusters. We demonstrate that our novel approach yields competitive and highly promising results on standard image clustering benchmarks. Finally, we showcase its robustness and the advantages of its improved interpretability by visualizing clustering results over real photograph collections.

## 1   Introduction

Gathering collections of images on a topic of interest is getting easier every day: simple tools can aggregate data from social media, web search, or specialized websites and filter it using hashtags, GPS coordinates, or semantic labels. However, identifying visual trends in such image collections remains difficult and usually involves manually organizing images or designing an ad hoc algorithm. Our goal in this paper is to design a clustering method which can be applied to such image collections, output a visual representation for each cluster and show how it relates to every associated image.

Directly comparing image pixels to decide if they belong to the same cluster leads to poor results because they are strongly impacted by factors irrelevant to clustering, such as exact viewpoint or lighting. Approaches to obtain clusters invariant to these transformations can be broadly classified into two groups. A first set of methods extracts invariant features and performs clustering in feature space. The features can be manually designed, but most state-of-the-art methods learn them directly from data. This is challenging because images are high-dimensional and learning relevant invariances thus requires huge amounts of data. For this reason, while recent approaches perform well on simple datasets like MNIST, they still struggle with real images. Another limitation of these approaches is that learned features are hard to interpret and visualize, making clustering results difficult to analyze. A second set of approaches, following the seminal work of Frey and Jojic on transformation-invariant clustering [11, 12, 13], uses explicit transformation models to align images before comparing them. These approaches have several potential advantages: (i) they enable direct control of the invariances to consider; (ii) because they do not need to discover invariances, they are potentially less data-hungry; (iii) since images are explicitly aligned, clustering process and results can easily be visualized. However, transformation-invariant approaches require solving a difficult joint optimization problem. In practice, they are thus often limited to small datasets and simple transformations, such as affine

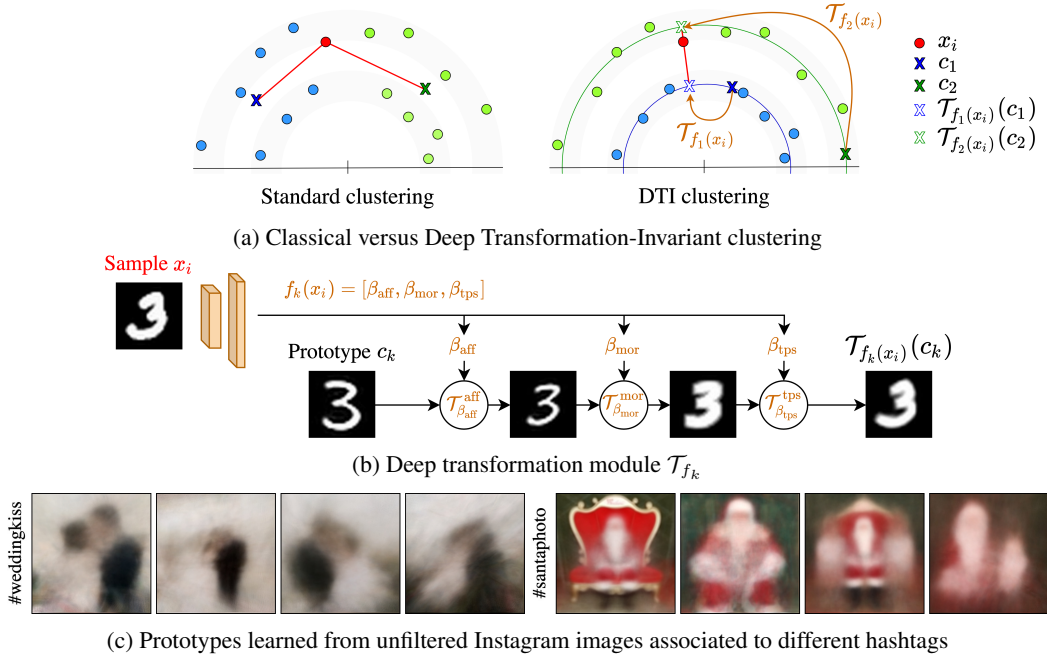

(a) Classical versus Deep Transformation-Invariant clustering

(b) Deep transformation module $\mathcal{T}_{f_k}$

(c) Prototypes learned from unfiltered Instagram images associated to different hashtags

Figure 1: **Overview.** **(a)** Given a sample $x_i$ and prototypes $c_1$ and $c_2$, standard clustering such as K-means assigns the sample to the closest prototype. Our DTI clustering first aligns prototypes to the sample using a family of parametric transformations - here rotations - then picks the prototype whose alignment yields the smallest distance. **(b)** We predict alignment with deep learning. Given an image $x_i$, each parameter predictor $f_k$ predicts parameters for a sequence of transformations - here affine $\mathcal{T}_{\beta_{\mathrm{aff}}}^{\mathrm{aff}}$, morphological $\mathcal{T}_{\beta_{\mathrm{mor}}}^{\mathrm{mor}}$, and thin plate spline $\mathcal{T}_{\beta_{\mathrm{tps}}}^{\mathrm{tps}}$ - to align prototype $c_k$ to $x_i$. **(c)** Examples of interpretable prototypes discovered from large images sets (15k each) associated to hashtags in Instagram using our DTI clustering with 40 clusters. Each cluster contains from 200 to 800 images.

transformations, and to the best of our knowledge they have never been evaluated on large standard image clustering datasets.

In this paper, we propose a deep transformation-invariant (DTI) framework that enables to perform transformation-invariant clustering at scale and uses complex transformations. Our main insight is to jointly learn deep alignment and clustering parameters with a single loss, relying on the gradient-based adaptations of K-means [38] and GMM optimization [9]. Not only is predicting transformations more computationally efficient than optimizing them, but it enables us to use complex color, thin plate spline and morphological transformations without any specific regularization. Because it is pixel-based, our deep transformation-invariant clustering is also easy to interpret: cluster centers and image alignments can be visualized to understand assignments. Despite its apparent simplicity, we demonstrate that our DTI clustering framework leads to results on par with the most recent feature learning approaches on standard benchmarks. We also show it is capable of discovering meaningful modes in real photograph collections, which we see as an important step to bridge the gap between theoretically well-grounded clustering approaches and semi-automatic tools relying on hand-designed features for exploring image collections, such as AverageExplorer [52] or ShadowDraw [32].

We first briefly discuss related works in Section 2. Section 3 then presents our DTI framework (Fig. 1a). Section 4 introduces our deep transformation modules and architecture (Fig. 1b) and discuss training details. Finally, Section 5 presents and analyzes our results (Fig. 1c).

**Contributions.** In this paper we present:
- a deep transformation-invariant clustering approach that jointly learns to cluster and align images,
- a deep image transformation module to learn spatial alignment, color modifications and for the first time morphological transformations,
- an experimental evaluation showing that our approach is competitive on standard image clustering benchmarks, improving over state-of-the-art on Fashion-MNIST and SVHN, and provides highly interpretable qualitative results even on challenging web image collections.

Code, data, models as well as more visual results are available on our project webpage[1].

## 2 Related work

Most recent approaches to image clustering focus on learning deep image representations, or features, on which clustering can be performed. Common strategies include autoencoders [48, 10, 25, 28], contrastive approaches [49, 5, 44], GANs [6, 51, 41] and mutual information based strategies [22, 18, 24]. Especially related to our work is [28] which leverages the idea of capsule [20] to learn equivariant image features, in a similar fashion of equivariant models [33, 45]. However, our method aims at being invariant to transformations but not at learning a representation.

Another type of approach is to align images in pixel space using a relevant family of transformations, such as translations, rotations, or affine transformations to obtain more meaningful pixel distances before clustering them. Frey and Jojic first introduced transformation-invariant clustering [11, 12, 13] by integrating pixel permutations as a discrete latent variable within an Expectation Maximization (EM) [9] procedure for a mixture of Gaussians. Their approach was however limited to a finite set of discrete transformations. *Congealing* generalized the idea to continuous parametric transformations, and in particular affine transformations, initially by using entropy minimization [40, 30]. A later version using least square costs [7, 8] demonstrated the relation of this approach to the classical Lukas-Kanade image alignment algorithm [37]. In its classical version, congealing only enables to align all dataset images together, but the idea was extended to clustering [36, 39, 34], for example using a Bayesian model [39], or in a spectral clustering framework [34]. These works typically formulate difficult joint optimization problems and solve them by alternating between clustering and transformation optimization for each sample. They are thus limited to relatively small datasets and to the best of our knowledge were never compared to modern deep approaches on large benchmarks. Deep learning was recently used to scale the idea of congealing for global alignment of a single class of images [1] or time series [46]. Both works build on the idea of Spatial Transformer Networks [23] (STN) that spatial transformation are differentiable and can be learned by deep networks. We also build upon STN, but go beyond single-class alignment to jointly perform clustering. Additionally, we extend the idea to color and morphological transformations. We believe our work is the first to use deep learning to perform clustering in pixel space by explicitly aligning images.

## 3 Deep Transformation-Invariant clustering

In this section, we first discuss a generic formulation of our deep transformation-invariant clustering approach, then derive two algorithms based on K-means [38] and Gaussian mixture model [9].

**Notation:** In all the rest of the paper, we use the notation $a_{1:n}$ to refer to the set $\{a_1, \ldots, a_n\}$.

### 3.1 DTI framework

Contrary to most recent image clustering methods which rely on feature learning, we propose to perform clustering in pixel space by making the clustering invariant to a family of transformations. We consider $N$ image samples $x_{1:N}$ and aim at grouping them in $K$ clusters using a *prototype method*. More specifically, each cluster $k$ is defined by a prototype $c_k$, which can also be seen as an image, and prototypes are optimized to minimize a loss $\mathcal{L}$ which typically evaluates how well they represent the samples. We further assume that $\mathcal{L}$ can be written as a sum of a loss $l$ computed over each sample:

$$\mathcal{L}(c_{1:K}) = \sum_{i=1}^{N} l(x_i, \{c_1, \ldots, c_K\}). \tag{1}$$

Once the problem is solved, each sample $x_i$ will be associated to the closest prototype.

Our key assumption is that in addition to the data, we have access to a group of parametric transformations $\{\mathcal{T}_\beta, \beta \in B\}$ to which we want to make the clustering invariant. For example, one can consider $\beta \in \mathbb{R}^6$ and $\mathcal{T}_\beta$ the 2D affine transformation parametrized by $\beta$. Other transformations are discussed in Section 4.1. Instead of finding clusters by minimizing the loss of Equation 1, one can

minimize the following transformation-invariant loss:

$$\mathcal{L}_{\text{TI}}(c_{1:K}) = \sum_{i=1}^{N} \min_{\beta_{1:K}} l(x_i, \{\mathcal{T}_{\beta_1}(c_1), \ldots, \mathcal{T}_{\beta_K}(c_K)\}). \qquad (2)$$

In this equation, the minimum over $\beta_{1:K}$ is taken for each sample independently. This loss is invariant to transformations of the prototypes (see proof in Appendix B). Also note there is not a single optimum since the loss is the same if any prototype $c_k$ is replaced by $\mathcal{T}_\beta(c_k)$ for any $\beta \in B$. If necessary, for example for visualization purposes, this ambiguity can easily be resolved by adding a small regularization on the transformations. The optimization problem associated to $\mathcal{L}_{\text{TI}}$ is of course difficult. A natural approach, which we use as baseline (noted TI), is to alternatively minimize over transformations and clustering parameters. We show that performing such optimization using a gradient descent can already lead to improved results over standard clustering but is computationally expensive.

We experimentally show it is faster and actually better to instead learn $K$ (deep) predictors $f_{1:K}$ for each prototype, which aim at associating to each sample $x_i$ the transformation parameters $f_{1:K}(x_i)$ minimizing the loss, i.e. to minimize the following loss:

$$\mathcal{L}_{\text{DTI}}(c_{1:K}, f_{1:K}) = \sum_{i=1}^{N} l(x_i, \{\mathcal{T}_{f_1(x_i)}(c_1), \ldots, \mathcal{T}_{f_K(x_i)}(c_K)\}), \qquad (3)$$

where predictors $f_{1:K}$ are now shared for all samples. We found that using deep parameters predictors not only enables more efficient training but also leads to better clustering results especially with more complex transformations. Indeed, the structure and optimization of the predictors naturally regularize the parameters for each sample, without requiring any specific regularization loss, especially in the case of high numbers $N$ of samples and transformation parameters.

In the next section we present concrete losses and algorithms. We then describe differentiable modules for relevant transformations and discuss parameter predictor architecture as well as training in Section 4.

## 3.2 Application to K-means and GMM

**K-means.** The goal of K-means algorithm [38] is to find a set of prototypes $c_{1:K}$ such that the average Euclidean distance between each sample and the closest prototype is minimized. Following the reasoning of Section 3.1, the loss optimized in K-means can be transformed into a transformation-invariant loss:

$$\mathcal{L}_{\text{DTI K-means}}(c_{1:K}, f_{1:K}) = \sum_{i=1}^{N} \min_k \|x_i - \mathcal{T}_{f_k(x_i)}(c_k)\|^2. \qquad (4)$$

Following batch gradient-based trainings [3] of K-means, we can then simply jointly minimize $\mathcal{L}_{\text{DTI K-means}}$ over prototypes $c_{1:K}$ and deep transformation parameter predictors $f_{1:K}$ using a batch gradient descent algorithm. In practice, we initialize prototypes $c_{1:K}$ with random samples and predictors $f_{1:K}$ such that $\forall k, \forall x, \mathcal{T}_{f_k(x)} = \text{Id}$.

**Gaussian mixture model.** We now consider that data are observations of a mixture of $K$ multivariate normal random variables $X_{1:K}$, i.e. $X = \sum_k \delta_{k,\Delta} X_k$ where $\delta$ is the Kronecker function and $\Delta \in \{1, \ldots, K\}$ is a random variable defined by $P(\Delta = k) = \pi_k$, with $\forall k, \pi_k > 0$ and $\sum_k \pi_k = 1$. We write $\mu_k$ and $\Sigma_k$ the mean and covariance of $X_k$ and $G(\,\textbf{.}\,; \mu_k, \Sigma_k)$ associated probability density function. The transformation-invariant negative log-likelihood can then be written:

$$\mathcal{L}_{\text{DTI GMM}}(\mu_{1:K}, \Sigma_{1:K}, \pi_{1:K}, f_{1:K}) = -\sum_{i=1}^{N} \log \Big( \sum_{k=1}^{K} \pi_k G\big(x_i\,; \mathcal{T}_{f_k(x_i)}(\mu_k), \mathcal{T}^*_{f_k(x_i)}(\Sigma_k)\big)\Big), \quad (5)$$

where $\mathcal{T}^*$ is slightly modified version of $\mathcal{T}$. Indeed, $\mathcal{T}$ may include transformations that one can apply to the covariance, such as spatial transformations, and other that would not make sense, such as additive color transformations. We jointly minimize $\mathcal{L}_{\text{DTI GMM}}$ over Gaussian parameters, mixing probabilities, and deep transformation parameters $f_{1:K}$ using a batch gradient-based EM procedure similar to [21, 15, 14] and detailed in Algorithm 1. In practice, we assume that pixels are independent resulting in diagonal covariance matrices.

In such gradient-based procedures, two constraints have to be enforced, namely the positivity and normalization of mixing probabilities $\pi_k$ and the non-negativeness of the diagonal covariance terms.

**Algorithm 1:** Deep Transformation-Invariant Gaussian Mixture Model

**Input:** data $\mathbf{X}$, number of clusters $K$, transformation $\mathcal{T}$
**Output:** cluster assignments, Gaussian parameters $\mu_{1:K}, \Sigma_{1:K}$, deep predictors $f_{1:K}$
**Initialization:** $\mu_{1:K}$ with random samples, $\Sigma_{1:K} = 0.5$, $\eta_{1:K} = 1$ and $\forall k, \forall x, \mathcal{T}_{f_k(x)} = \mathrm{Id}$
**while** *not converged* **do**

     i. sample a batch of data points $x_{1:N}$
     ii. compute mixing probabilities:    $\pi_{1:K} = \mathrm{softmax}(\eta_{1:K})$
     iii. compute per-sample Gaussian transformed parameters:

$$\forall k, \ \forall i, \ \ \tilde{\mu}_{ki} = \mathcal{T}_{f_k(x_i)}(\mu_k) \ \ \text{and} \ \ \tilde{\Sigma}_{ki} = \mathcal{T}^*_{f_k(x_i)}(\Sigma_k) + \mathrm{diag}(\sigma^2_{\min})$$

     iv. compute responsibilities: $\forall k, \ \forall i, \ \gamma_{ki} = \frac{\pi_k G(x_i\,;\tilde{\mu}_{ki},\tilde{\Sigma}_{ki})}{\sum_j \pi_j G(x_i\,;\tilde{\mu}_{ji},\tilde{\Sigma}_{ji})}$            (*E-step*)

     v. minimize expected negative log-likelihood w.r.t to $\{\mu_{1:K}, \Sigma_{1:K}, \eta_{1:K}, f_{1:K}\}$:

$$\mathbb{E}[\mathcal{L}_{\mathrm{DTI\,GMM}}] = -\sum_{i=1}^{N}\sum_{k=1}^{K} \gamma_{ki}\Big(\log\big(G(x_i\,;\tilde{\mu}_{ki},\tilde{\Sigma}_{ki})\big) + \log(\pi_k)\Big) \qquad (\textit{M-step})$$

**end**

For the mixing probabilities constraints, we adopt the approach used in [21] and [14] which optimize mixing parameters $\eta_k$ used to compute the probabilities $\pi_k$ using a softmax instead of directly optimizing $\pi_k$, which we write $\pi_{1:K} = \mathrm{softmax}(\eta_{1:K})$. For the variance non-negativeness, we introduce a fixed minimal variance value $\sigma^2_{\min}$ which is added to the variances when evaluating the probability density function. This approach is different from the one in [14] which instead use clipping, because we found training with clipped values was harder. In practice, we take $\sigma_{\min} = 0.25$.

## 4 Learning image transformations

### 4.1 Architecture and transformation modules

We consider a set of prototypes $c_{1:K}$ we would like to transform to match a given sample $x$. To do so, we propose to learn for each prototype $c_k$, a separate deep predictor which predicts transformation parameters $\beta$. We propose to model the family of transformations $\mathcal{T}_\beta$ as a sequence of M parametric transformations such that, writing $\beta = (\beta^1, \ldots, \beta^M)$, $\mathcal{T}_\beta = \mathcal{T}^M_{\beta^M} \circ \ldots \circ \mathcal{T}^1_{\beta^1}$. In the following, we describe the architecture of transformation parameter predictors $f_{1:K}$, as well as each family of parametric transformation modules we use. Figure 1b shows our learned transformation process on a MNIST example.

**Parameters prediction network.** For all experiments, we use the same parameter predictor network architecture composed of a shared ResNet [19] backbone truncated after the global average pooling, followed by $K \times M$ Multi-Layer Perceptrons (MLPs), one for each prototype and each transformation module. For the ResNet backbone, we use ResNet-20 for images smaller than $64 \times 64$ and ResNet-18 otherwise. Each MLP has the same architecture, with two hidden layers of 128 units.

**Spatial transformer module.** To model spatial transformations of the prototypes, we follow the spatial transformers developed by Jaderberg et al. [23]. The key idea is to model spatial transformations as a differentiable image sampling of the input using a deformed sampling grid. We use affine $\mathcal{T}^{\mathrm{aff}}_\beta$, projective $\mathcal{T}^{\mathrm{proj}}_\beta$ and thin plate spline $\mathcal{T}^{\mathrm{tps}}_\beta$ [2] transformations which respectively correspond to 6, 8 and 16 (a 4x4 grid of control points) parameters.

**Color transformation module.** We model color transformation with a channel-wise diagonal affine transformation on the full image, which we write $\mathcal{T}^{\mathrm{col}}_\beta$. It has 2 parameters for greyscale images and 6 parameters for colored images. We first used a full affine transformation with 12 parameters, however the network was able to hide several patterns in the different color channels of a single prototype (Appendix C.4). Note that a similar transformation was theoretically introduced in capsules [28], but with the different goal of obtaining a color-invariant feature representation. Deep feature-based approaches often handle color images with a pre-processing step such as Sobel filtering [4, 24, 28]. We believe the way we align colors of the prototypes to obtain color invariance in pixel space is novel, and it enables us to directly work with colored images without using any pre-processing or specific invariant features.

**Morphological transformation module.** We introduce a new transformation module to learn morphological operations [16] such as dilation and erosion. We consider a greyscale image $x \in \mathbb{R}^D$ of size $U \times V = D$, we write $x[u, v]$ the value of the pixel $(u, v)$ for $u \in \{1, \dots, U\}$ and $v \in \{1, \dots, V\}$. Given a 2D region $A$, the dilation of $x$ by $A$, $\mathcal{D}_A(x) \in \mathbb{R}^D$, is defined by $\mathcal{D}_A(x)[u, v] = \max_{(u', v') \in A} x[u + u', v + v']$ and its erosion by $A$, $\mathcal{E}_A(x) \in \mathbb{R}^D$, is defined by $\mathcal{E}_A(x)[u, v] = \min_{(u', v') \in A} x[u + u', v + v']$. Directly learning the region $A$ which parametrizes these transformations is challenging, we thus propose to learn parameters $(\alpha, a)$ for the following soft version of these transformations:

$$\mathcal{T}_{(\alpha, a)}^{\text{mor}}(x)[u, v] = \frac{\sum_{(u', v') \in W} x[u + u', v + v'] \cdot a[u + u', v + v'] \cdot e^{\alpha x[u + u', v + v']}}{\sum_{(u', v') \in W} a[u + u', v + v'] \cdot e^{\alpha x[u + u', v + v']}}, \quad (6)$$

where $W$ is a fixed set of 2D positions, $\alpha$ is a softmax (positive values) or softmin (negative values) parameter and $a$ is a set of parameters with values between 0 and 1 defined for every position $(u', v') \in W$. Parameters $a$ can be interpreted as an image, or as a soft version of the region $A$ used for morphological operations. Note that if $a[u', v'] = \mathbf{1}_{\{(u', v') \in A\}}$, when $\alpha \to +\infty$ (resp. $-\infty$), it successfully emulates $\mathcal{D}_A$ (resp. $\mathcal{E}_A$). In practice, we use a grid of integer positions around the origin of size $7 \times 7$ for $W$. Note that since morphological transformations do not form a group, transformation-invariant denomination is slightly abusive.

## 4.2 Training

We found that two key elements were critical to obtain good results: empty cluster reassignment and curriculum learning. We then discuss further implementation details and computational cost.

**Empty cluster reassignment.** Similar to [4], we adopt an empty cluster reassignment strategy during our clustering optimization. We reinitialize both prototype and deep predictor of "tiny" clusters using the parameters of the largest cluster with a small added noise. In practice, the size of balanced clusters being $N/K$, we define "tiny" as less than 20% of $N/K$.

**Curriculum learning.** Learning to predict transformations is a hard task, especially when the number of parameters is high. To ease learning, we thus adopt a curriculum learning strategy by gradually adding more complex transformation modules to the training. Given a target sequence of transformations to learn, we first train our model without any transformation - or equivalently with an identity module - then iteratively add subsequent modules once convergence has been reached. We found this is especially important when modeling local deformations with complex transformations with many parameters, such as TPS and morphological transformations. Intuitively, prototypes should first be coarsely aligned before attempting to refine the alignment with more complex transformations.

**Implementation details.** Both clustering parameters and parameter prediction networks are learned jointly and end-to-end using Adam optimizer [27] with a $10^{-6}$ weight decay on the neural network parameters. We sequentially add transformation modules at a constant learning rate of 0.001 then divide the learning rate by 10 after convergence - corresponding to different numbers of epochs depending on the dataset characteristics - and train for a few more epochs with the smaller learning rate. We use a batch size of 64 for real photograph collections and 128 otherwise.

**Computational cost.** Training DTI K-means or DTI GMM on MNIST takes approximately 50 minutes on a single Nvidia GeForce RTX 2080 Ti GPU and full dataset inference takes 30 seconds. We found it to be much faster than directly optimizing transformation parameters (TI clustering) for which convergence took more than 10 hours of training.

## 5 Experiments

In this section, we first analyze our approach and compare it to state-of-the-art, then showcase its interest for image collection analysis and visualization.

### 5.1 Analysis and comparisons

Similar to previous work on image clustering, we evaluate our approach with global classification accuracy (ACC), where a cluster-to-class mapping is computed using the Hungarian algorithm [29],

Table 1: **Comparisons.** We report ACC and NMI in % on standard clustering benchmarks. Symbols mark methods that use data augmentation (▽) and manually selected features as input (§ for pretrained features from best VaDE run, † for GIST features, ‡ for Sobel filters) and are thus not directly comparable. For SVHN, we also report our results with our Gaussian weighted loss (⋆). Eval column refers to the aggregate used: best run (*max*), average (*avg*) or run with minimal loss (*minLoss*).

| Method | Runs | Eval | MNIST ACC | MNIST NMI | MNIST-test ACC | MNIST-test NMI | USPS ACC | USPS NMI | F-MNIST ACC | F-MNIST NMI | FRGC ACC | FRGC NMI | SVHN ACC |
|---|---|---|---|---|---|---|---|---|---|---|---|---|---|
| *Clustering on a learned feature* | | | | | | | | | | | | | |
| DEC [48, 50] | 9 | max | 86.3 | 83.4 | 85.6 | 83.0 | 76.2 | 76.7 | 51.8 | 54.6 | 37.8 | 50.5 | - |
| InfoGAN [6, 41] | 5 | max | 89.0 | 86.0 | - | - | - | - | 61.0 | 59.0 | - | - | - |
| VaDE [25, 50] | 10 | max | 94.5 | 87.6 | - | - | 56.6 | 51.2 | 57.8 | 63.0 | - | - | - |
| ClusterGAN [41] | 5 | max | 95.0 | 89.0 | - | - | - | - | 63.0 | 64.0 | - | - | - |
| JULE [49] | 3 | avg | 96.4 | 91.3 | 96.1 | 91.5 | 95.0 | 91.3 | 56.3 | 60.8 | 46.1 | 57.4 | - |
| DEPICT [10] | 5 | avg | 96.5 | 91.7 | 96.3 | 91.5 | **96.4** | **92.7** | 39.2 | 39.2 | **47.0** | **61.0** | - |
| DSCDAN [50] | 10 | avg | **97.8** | **94.1** | **98.0** | **94.6** | 86.9 | 85.7 | **66.2** | **64.5** | - | - | - |
| *Clustering on a learned feature with data augmentation and/or ad hoc data representation* | | | | | | | | | | | | | |
| SpectralNet [44] | 5 | avg | 97.1§ | 92.4§ | - | - | - | - | - | - | - | - | - |
| IMSAT [22] | 12 | avg | 98.4▽ | - | - | - | - | - | - | - | - | - | **57.3**▽† |
| ADC [18] | 20 | avg | 98.7▽ | - | - | - | - | - | - | - | 43.7▽ | - | 38.6▽ |
| SCAE [28] | 5 | avg | 98.7▽ | - | - | - | - | - | - | - | - | - | 55.3‡ |
| IIC [24] | 5 | avg | 98.4▽ | - | - | - | - | - | - | - | - | - | - |
| | 5 | minLoss | **99.2**▽ | - | - | - | - | - | - | - | - | - | - |
| *Clustering on pixel values* | | | | | | | | | | | | | |
| K-means [38] | 10 | avg | 54.8 | 50.2 | 55.9 | 51.2 | 65.3 | 61.2 | 54.1 | 51.4 | 22.7 | 26.5 | 12.2 |
| GMM [9] | 10 | avg | 54.2 | 51.7 | 55.6 | 54.7 | 66.0 | 60.9 | 49.7 | 51.2 | 24.2 | 27.9 | 11.6 |
| **DTI K-means** | 10 | avg | **97.3** | **94.0** | 96.6 | 94.6 | 86.4 | 88.2 | 61.2 | 63.7 | 39.6 | 48.7 | 36.4 / 44.5⋆ |
| | 10 | minLoss | 97.2 | 93.8 | **98.0** | **95.3** | **89.8** | **89.5** | 57.4 | 64.1 | 49.7 | 39.6 / 62.6⋆ |
| **DTI GMM** | 10 | avg | 95.9 | 93.2 | 97.8 | 94.7 | 84.5 | 87.2 | 59.6 | 62.2 | 40.1 | 48.9 | 36.7 / 57.4⋆ |
| | 10 | minLoss | 97.1 | 93.7 | **98.0** | 95.1 | 87.3 | 89.0 | **68.2** | **66.3** | 41.6 | 51.1 | 39.5 / **63.3**⋆ |

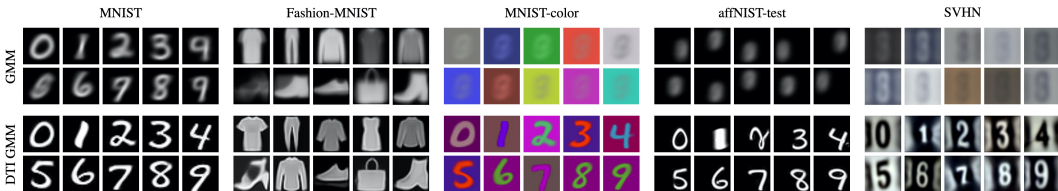

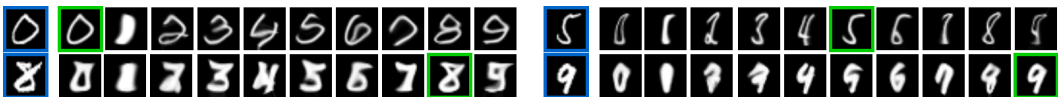

(a) Prototypes learned for different datasets

(b) Transformations predicted for all prototypes for 4 MNIST images

Figure 2: **Qualitative results. (a)** compares prototypes learned from GMM and our DTI GMM, **(b)** shows transformed prototypes given query samples from MNIST and highlight the closest prototype.

and Normalized Mutual Information (NMI). Datasets and corresponding transformation modules we used are described in Appendix A.

**Comparison on standard benchmarks.** In Table 1, we report our results on standard image clustering benchmarks, i.e. digit datasets (MNIST [31], USPS [17]), a clothing dataset (Fashion-MNIST [47]) and a face dataset (FRGC [43]). We also report results for SVHN [42] where concurrent methods use pre-processing to remove color bias. In the table, we separate representation-based from pixel-based methods and mark results using data augmentation or manually selected features as input. Note that our results depend on initialization, we provide detailed statistics in Appendix C.1.

Our DTI clustering is fully unsupervised and does not require any data augmentation, ad hoc features, nor any hyper-parameter while performing clustering directly in pixel space. We report average performances and performances of the minimal loss run which we found to correlate well with high performances (Appendix C.2). Because this non-trivial criterion allows to automatically select a run in a fully unsupervised way, we argue it can be compared to average results from competing methods which don't provide such criterion.

First, DTI clustering achieves competitive results on all datasets, in particular improving state-of-the-art by a significant margin on SVHN and Fashion-MNIST. For SVHN, we first found that the

Table 2: **Augmented and specific datasets.** Clustering accuracy (%) with standard deviation for methods applied on raw images (no pre-processing). We used 10 runs for our method and 5 for the baselines.

| Method | Eval | MNIST-1k | MNIST-color | affNIST-test |
|---|---|---|---|---|
| VaDE [25] | avg | 49.6 (5.6) | 11.9 (1.2) | Div. |
| IMSAT [22] | avg | 67.9 (2.3) | 10.6 (0.1) | 18.2 (2.6) |
| IIC [24] | avg | 63.4 (0.4) | 10.6 (0.0) | 57.6 (0.0) |
| | minLoss | 63.2 | 10.6 | 57.6 |
| **DTI K-means** | avg | 79.8 (6.9) | 96.7 (0.1) | 95.5 (3.3) |
| | minLoss | **90.5** | **96.8** | **97.0** |
| **DTI GMM** | avg | 80.8 (7.2) | 96.0 (0.2) | 93.3 (5.9) |
| | minLoss | 87.1 | 95.8 | **97.0** |

Table 3: **Ablation study on MNIST.** Clustering accuracy (%) over 10 runs.

| Method | Avg | MinLoss |
|---|---|---|
| **DTI clustering (aff-morpho-tps)** | **97.3** | **97.2** |
| ordering: aff-tps-morpho | 95.5 | 96.9 |
| ordering: morpho-aff-tps | 27.5 | 97.0 |
| w/o morphological | 94.8 | 95.8 |
| w/o thin plate spline | 90.0 | 82.5 |
| w/o affine | 85.1 | 96.8 |
| affine only | 90.1 | 90.5 |
| w/o empty cluster reassignment | 80.9 | 78.6 |
| w/o curriculum learning | 83.9 | 78.9 |
| TI clustering (aff-morpho-tps, 1 run) | 26.3 | 26.3 |
| TI clustering (affine only) | 73.0 | 73.1 |

prototypes quality was harmed by digits on the side of the image. To pay more attention to the center digit, we weighted the clustering loss by a Gaussian weight ($\sigma = 7$). It led to better prototypes and allowed us to improve over all concurrent methods by a large margin. Compared to representation-based methods, our pixel-based clustering is highly interpretable. Figure 2a shows standard GMM prototypes and our prototypes learned with DTI GMM which appear to be much sharper than standard ones. This directly stems from the quality of the learned transformations, visualized in Figure 2b. Our transformation modules can successfully align the prototype, adapt the thickness and apply local elastic deformations. More alignment results are available on our project webpage.

**Augmented and specific datasets.** DTI clustering also works on small, colored and misaligned datasets. In Table 2, we highlight these strengths on specifics datasets generated from MNIST: MNIST-1k is a 1000 images subset, MNIST-color is obtained by randomly selecting a color for the foreground and background and affNIST-test[2] is the result of random affine transformations. We used an online implementation[3] for VaDE [25] and official ones for IMSAT [22] and IIC [24] to obtain baselines. Our results show that the performances of DTI clustering is barely affected by spatial and color transformations, while baseline performances drop on affNIST-test and are almost chance on MNIST-color. Figure 2a shows the quality and interpretability of our cluster centers on affNIST-test and MNIST-color. DTI clustering also seems more data-efficient than the baselines we tested.

**Ablation on MNIST.** In Table 3, we conduct an ablation study on MNIST of our full model trained following Section 4.2 with affine, morphological and TPS transformations. We first explore the effect of transformation modules. Their order is not crucial, as shown by similar minLoss performances, but can greatly affect the stability of the training, as can be seen in the average results. Each module contributes to the final performance, affine transformations being the most important. We then validate our training strategy showing that both empty cluster reassignment and curriculum learning for the different modules are necessary. Finally, we directly optimize the loss of Equation 2 (TI clustering) by optimizing the transformation parameters for each sample at each iteration of the batch clustering algorithm, without using our parameter predictors. With rich transformations which have many parameters, such as TPS and morphological ones, this approach fails completely. Using only affine transformations, we obtain results clearly superior to standard clustering, but worse than ours.

## 5.2 Application to web images

One of the main interest of our DTI clustering is that it allows to discover trends in real image collections. All images are resized and center cropped to $128 \times 128$. The selection of the number of clusters is a difficult problem and is discussed in Appendix C.3.

In Figure 1c, we show examples of prototypes discovered in very large unfiltered sets (15k each) of Instagram images associated to different hashtags[4] using DTI GMM applied with 40 clusters. While many images are noise and are associated to prototypes which are not easily interpretable, we show prototypes where iconic photos and poses can be clearly identified. To the best of our knowledge, we believe we are the first to demonstrate this type of results from raw social network image collections.

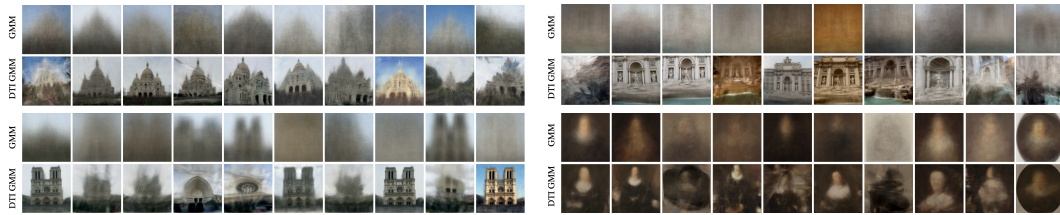

(a) Full sets of prototypes discovered with GMM and DTI GMM

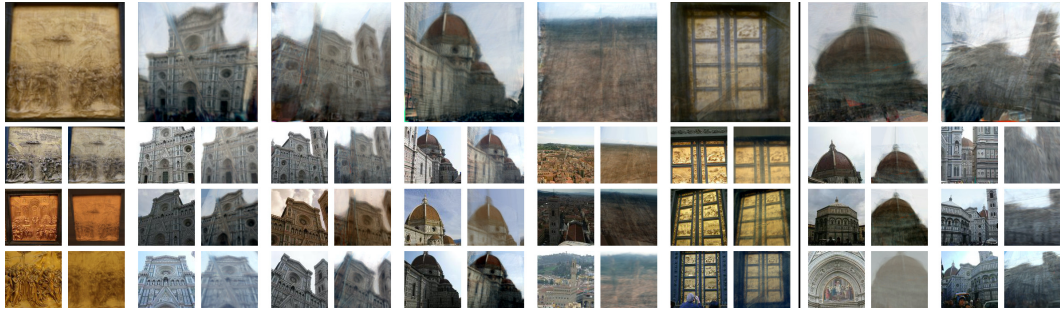

(b) Examples of cluster centers and aligned images with DTI GMM (20 clusters)

Figure 3: **Qualitative results on real photographs. (a)** Clustering results from photographs of different locations in [35] (1,089 Sacre Coeur top-left, 1,688 Trevi fountain top-right, 2,625 Notre-Dame bottom-left) and 980 Baroque portraits from [26] (bottom-right). **(b)** Clustering results from 1,892 Florence cathedral images from [35]. Top row shows learned prototypes while the three bottom rows show examples of images from each cluster and aligned prototypes. These clusters contain respectively 44, 154, 134, 64, 71, 133, 85 and 64 images. The left six examples are successful clusters while the two right clusters are relative failure cases.

Comparable results in AverageExplorer [52], e.g. on Santa images, could be obtained using ad hoc features and user interactions, while our results are produced fully automatically.

Figure 3 shows qualitative clustering results on MegaDepth [35] and WikiPaintings [26]. Similar to our results on image clustering benchmarks, our learned prototypes are more relevant and accurate than the ones obtained from standard clustering. Note that some of our prototypes are very sharp: they typically correspond to sets of photographs between which we can accurately model deformations, e.g. scenes that are mostly planar, with little perspective effects. On the contrary, more unique photographs and photographs with strong 3D effects that we cannot model will be associated to less interpretable and blurrier prototypes, such as the ones in the last two columns of Figure 3b. In Figure 3b, in addition to the prototypes discovered, we show examples of images contained in each cluster as well as the aligned prototype. Even for such complex images, the simple combination of our color and spatial modules manages to model real image transformations like illumination variations and viewpoint changes. More web image clustering results are shown on our project webpage.

## 6 Conclusion

We have introduced an efficient deep transformation-invariant clustering approach in raw input space. Our key insight is the online optimization of a single clustering objective over clustering parameters and deep image transformation modules. We demonstrate competitive results on standard image clustering benchmarks, including improvements over state-of-the-art on SVHN and Fashion-MNIST. We also demonstrate promising results for real photograph collection clustering and visualization. Finally, note that our DTI clustering framework is not specific to images and can be extended to other types of data as long as appropriate transformation modules are designed beforehand.

## Acknowledgements

This work was supported in part by ANR project EnHerit ANR-17-CE23-0008, project Rapid Tabasco, gifts from Adobe and HPC resources from GENCI-IDRIS (Grant 2020-AD011011697). We thank Bryan Russell, Vladimir Kim, Matthew Fisher, François Darmon, Simon Roburin, David Picard, Michaël Ramamonjisoa, Vincent Lepetit, Elliot Vincent, Jean Ponce, William Peebles and Alexei Efros for inspiring discussions and valuable feedback.

## Broader Impact

The impact of clustering mainly depends on the data it is applied on. For instance, adding structure in user data can raise ethical concerns when users are assimilated to their cluster, and receive targeted advertisement and newsfeed. However, this is not specific to our method and can be said of any clustering algorithm. Also note that while our clustering can be applied for example to data from social media, the visual interpretation of the clusters it returns via the cluster centers respects privacy much better than showing specific examples from each cluster.

Because our method provides highly interpretable results, it might bring increased understanding of clustering algorithm results for the broader public, which we think may be a significant positive impact.

## Footnotes

[1]http://imagine.enpc.fr/~monniert/DTIClustering/

[2]https://www.cs.toronto.edu/~tijmen/affNIST/

[3]https://github.com/GuHongyang/VaDE-pytorch

[4]https://github.com/arc298/instagram-scraper was used to scrape photographs

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
