[Supplementary Material]

# A Dataset descriptions

Table 4: **Datasets and transformation sequences used**

| Dataset | Samples | Classes | Dimension | Transformation sequence |
|---|---|---|---|---|
| *Standard* | | | | |
| MNIST [31] | 70,000 | 10 | $1\times28\times28$ | aff-morpho-tps |
| MNIST-test [31] | 10,000 | 10 | $1\times28\times28$ | aff-morpho-tps |
| USPS [17] | 9,298 | 10 | $1\times16\times16$ | col-aff-tps |
| Fashion-MNIST [47] | 70,000 | 10 | $1\times28\times28$ | col-aff-tps |
| FRGC [43] | 2,462 | 20 | $3\times32\times32$ | col-aff-tps |
| SVHN [42] | 99,289 + unlabeled extra | 10 | $3\times28\times28$ | col-proj |
| *Augmented* | | | | |
| MNIST-1k | 1,000 | 10 | $1\times28\times28$ | aff-morpho-tps |
| MNIST-color | 70,000 | 10 | $3\times28\times28$ | col-aff-tps |
| affNIST-test | 320,000 | 10 | $1\times40\times40$ | aff-morpho-tps |
| *Real photographs* | | | | |
| All | 1k to 15k | - | $3\times128\times128$ | col-proj |

Table 4 summarizes dataset characteristics as well as the transformation sequences used. Datasets are:

- **MNIST** and **MNIST-test** [31] which respectively correspond to full and test subset of MNIST dataset. They depict binary white handwritten digits centered over a black background.
- **USPS** [17] is a handwritten digit dataset from USPS composed of greyscale images.
- **Fashion-MNIST** [47] is a 10-class clothing dataset composed of greyscale images of cloth over black background. Classes are: T-shirt, trouser, pullover, dress, coat, sandal, shirt, sneaker, bag, ankle boot.
- **FRGC** [43] is a colored face dataset. We use a subset of this dataset introduced in [49], where 20 subjects are selected and each image is cropped and resized to a constant size of $32\times32$.
- **SVHN** [42] is composed of digits extracted from house numbers cropped from Google Street View images. Following standard practice for clustering, we use both labeled samples (99,289) and unlabeled extra samples (~530k) for training and evaluate on the labeled subset only.
- **affNIST-test** is the test split of affNIST (https://www.cs.toronto.edu/ tijmen/affNIST/) an augmented dataset of MNIST where random affine transformations are applied.
- **MNIST-1k**: we randomly sampled 1,000 images from the test split of MNIST.
- **MNIST-color**: we augmented MNIST with random colors for background and foreground.

# B Transformation invariance

We consider $N$ image samples $x_{1:N}$, $K$ prototypes $c_{1:K}$ and a group of parametric transformations $\{\mathcal{T}_\beta,\ \beta \in B\}$. For $\beta_1, \beta_2 \in B$, we write $\beta_1\beta_2 \in B$ the element such that $\mathcal{T}_{\beta_1\beta_2} = \mathcal{T}_{\beta_1} \circ \mathcal{T}_{\beta_2}$. We have, for any $\alpha_1, \ldots, \alpha_K \in B$:

$$\mathcal{L}_{\text{TI}}(\{c_1, \ldots, c_K\}) = \mathcal{L}_{\text{TI}}(\{\mathcal{T}_{\alpha_1}(c_1), \ldots, \mathcal{T}_{\alpha_K}(c_K)\}).$$

Indeed:

$$\mathcal{L}_{\text{TI}}(\{\mathcal{T}_{\alpha_1}(c_1), \ldots, \mathcal{T}_{\alpha_K}(c_K)\}) = \sum_{i=1}^{N} \min_{\{\beta_1, \ldots, \beta_K\} \in B^K} l(x_i, \{\mathcal{T}_{\beta_1} \circ \mathcal{T}_{\alpha_1}(c_1), \ldots, \mathcal{T}_{\beta_K} \circ \mathcal{T}_{\alpha_K}(c_K)\})$$

$$= \sum_{i=1}^{N} \min_{\{\beta_1, \ldots, \beta_K\} \in B^K} l(x_i, \{\mathcal{T}_{\beta_1\alpha_1}(c_1), \ldots, \mathcal{T}_{\beta_K\alpha_K}(c_K)\})$$

$$= \sum_{i=1}^{N} \min_{\{\beta'_1, \ldots, \beta'_K\} \in B^K} l(x_i, \{\mathcal{T}_{\beta'_1}(c_1), \ldots, \mathcal{T}_{\beta'_K}(c_K)\})$$

$$= \mathcal{L}_{\text{TI}}(\{c_1, \ldots, c_K\}),$$

using the variable change $\beta'_k = \beta_k\alpha_k$, which is possible because for any $\alpha \in B$, $\alpha B = B$ as we assumed to have a group of transformations.

In some specific cases, the loss is also invariant to the samples, in particular when the loss $l$ is invariant to joint transformation of the prototype and the samples, i.e. for any $\beta \in B$, $l(x_i, \{c_1, \ldots, c_K\} = l(\mathcal{T}_\beta(x_i), \{\mathcal{T}_\beta(c_1), \ldots, \mathcal{T}_\beta(c_K)\})$. This is the case for example for K-means with a group of isometric transformations (e.g. rigid transformations), and it is also the case for GMM with the group of affine transformations applied to both the mean and covariance mixture parameters.

Note that we also tried to transform the samples to match the prototypes, which would lead to an invariance to sample transformation. However, a trivial solution to corresponding optimization problem is to learn "empty" prototypes and transformations of the samples into empty images. For examples, for the MNIST case with affine transformations, we observed that completely black prototypes were learned and any digit was transformed into a black image. Although a regularization term could have prevented such behaviour, we argue that keeping raw samples as target and transforming the prototypes is simpler and effective.

## C   Analysis

### C.1   Statistics on standard clustering benchmarks

Table 5: **Detailed results.** We report statistics of our results on standard clustering benchmarks. For SVHN, we also report results with our Gaussian weighted loss ($\star$).

| Method | Runs | Stat | MNIST ACC | NMI | MNIST-test ACC | NMI | USPS ACC | NMI | F-MNIST ACC | NMI | FRGC ACC | NMI | SVHN ACC |
|---|---|---|---|---|---|---|---|---|---|---|---|---|---|
| **DTI K-means** | 10 | avg | 97.3 | 94.0 | 96.6 | 94.6 | 86.4 | 88.2 | 61.2 | 63.7 | 39.6 | 48.7 | 36.4 / 44.5$^\star$ |
| | 10 | std | 0.1 | 0.1 | 4.1 | 1.5 | 4.1 | 1.6 | 2.0 | 0.3 | 1.7 | 2.2 | 1.9 / 9.6$^\star$ |
| | 10 | min | 97.1 | 93.8 | 84.9 | 90.4 | 83.2 | 87.1 | 57.4 | 63.2 | 35.9 | 43.9 | 34.5 / 37.0$^\star$ |
| | 10 | median | 97.3 | 94.0 | 97.9 | 95.1 | 85.0 | 87.4 | 61.9 | 63.3 | 40.2 | 49.3 | 35.8 / 39.6$^\star$ |
| | 10 | max | 97.5 | 94.2 | 98.0 | 95.3 | 96.4 | 92.0 | 63.3 | 64.2 | 41.1 | 51.4 | 39.6 / 62.6$^\star$ |
| | 10 | minLoss | 97.2 | 93.8 | 98.0 | 95.3 | 89.8 | 89.5 | 57.4 | 64.1 | 41.1 | 49.7 | 39.6 / 62.6$^\star$ |
| **DTI GMM** | 10 | avg | 95.9 | 93.2 | 97.8 | 94.7 | 84.5 | 87.2 | 59.6 | 62.2 | 40.1 | 48.9 | 36.7 / 57.4$^\star$ |
| | 10 | std | 3.9 | 1.5 | 0.1 | 0.2 | 2.0 | 0.8 | 4.7 | 2.4 | 1.4 | 1.5 | 2.3 / 5.1$^\star$ |
| | 10 | min | 84.7 | 89.1 | 97.7 | 94.4 | 82.0 | 86.3 | 56.1 | 59.7 | 38.4 | 46.8 | 34.0 / 49.9$^\star$ |
| | 10 | median | 97.1 | 93.7 | 97.8 | 94.7 | 84.3 | 87.1 | 57.2 | 60.9 | 39.6 | 49.1 | 36.4 / 57.4$^\star$ |
| | 10 | max | 97.3 | 93.9 | 98.0 | 95.1 | 87.3 | 89.0 | 68.2 | 66.3 | 41.9 | 51.1 | 39.5 / 64.6$^\star$ |
| | 10 | minLoss | 97.1 | 93.7 | 98.0 | 95.1 | 87.3 | 89.0 | 68.2 | 66.3 | 41.6 | 51.1 | 39.5 / 63.3$^\star$ |

### C.2   Accuracy and loss correlation

Similar to standard K-means and GMM, there is a variation in performances depending on the random initialization. We experimentally found that: (i) runs seem to be mainly grouped into distinct modes, each corresponding to roughly the same clustering quality; (ii) a run with a low loss usually leads to high clustering performances. We launched 100 runs on MNIST-test dataset and plot the loss with respect to the accuracy for each run in Figure 4. Except 2 outliers for the 100 runs, the runs with lower loss correspond to the runs with better performances. This is verified in most of our experiments, where the minLoss criterion clearly improves over the average performance.

Figure 4: **Accuracy/loss correlation.** We report loss and accuracy for DTI K-means on MNIST-test.

### C.3   Effect of the number of clusters K

Similar to many clustering methods, the selection of the number of clusters is a challenge. We investigated if a purely quantitative analysis could be applied to select $K$. In Figure 5a, we plot the loss of DTI-Kmeans as a function of the number of clusters for MNIST-test (left) and Notre-Dame (right). For MNIST-test, it is clear an elbow method could be applied to select the good number of clusters. For Notre-Dame, the quantitative analysis is not as conclusive but in this case, the correct number of clusters is not clearly defined. In practice, we did not find the qualitative results on internet photo collections to be very sensitive to this choice, as shown in Figure 5b where learned prototypes are mostly consistent across the different clustering results.

(a) Loss w.r.t varying numbers of clusters for MNIST-test (left) and Notre-Dame (right)

(b) Prototypes learned on Notre-Dame for different numbers of clusters

Figure 5: **Effect of** $K$**.** **(a)** We report the loss of DTI K-means for different numbers of clusters. For MNIST-test (left), the loss is averaged over 5 runs and for Notre-Dame (right), the loss corresponds to a single run. **(b)** We show the prototypes learned on Notre-Dame for even numbers of clusters.

## C.4 Constraining color transformation

While evaluating our approaches on real photograph collections, we experimentally observed that a full affine color transformation module (12 parameters) was too flexible and as a result, prototypes were able to learn different patterns hidden in each color channel. In Figure 6, we show each R, G and B channel as a greyscale image for two prototypes learned using a full affine color transformation module. One can see that a second pattern is hidden in particular in the green channels. To avoid this effect, we restricted the color transformation module to be a diagonal affine transformation corresponding to 6 parameters in total.

Figure 6: **Learned prototypes and RGB decomposition.** Two examples of learned prototypes (first column) on Florence cathedral collection from [35] using a full color transformation module (12 parameters). The 3 right columns correspond to R, G and B channels rescaled between 0 and 1. Note how the green channel is used to hide a completely different pattern from the other 2 channels.