[Reviews · NeurIPS 2020]

Review 1

Summary and Contributions: The paper presents an approach to image clustering, that computes the distance of each input image to a cluster prototype image, by first appropriately transforming the prototype to match the image and then computing the distance of the transformed protoype to the input image. The main idea is the training of deep neural networks (one for each transformation) that take an image as input and provide as output the corresponding transformation parameters.

Strengths: An image clustering approach is presented that joinly learns to cluster and align images. The method provides good empirical results on challenging web image collections.

Weaknesses: The main drawback of the paper is the lack of important details concerning the tranformation parameters that are predicted (section 4.1). No information is provided about the outputs of the networks that predict the transformation parmeters (e.g. how many are the parameters to be predicted). The optimization problem solved in the M-step seems to be hard. Performance depends on cluster initialization and network initialization. There is no comment on this issue.

Correctness: The method seems to be correct, however critical information is missing.

Clarity: Section 4.1 needs to be improved to provide information about the network outputs and the transformation parameters that are predicted.

Relation to Prior Work: Description of previous work is sufficient.

Reproducibility: No

Additional Feedback: The paper presents an approach to image clustering that jointly learns to cluster and align images. An interesting aspect of the paper is the application of the method to real photograph collections. The method relies on training deep neural networks that provide as outputs appropriate transformation parameters for each image. No information is provided about the outputs of the networks that predict the transformation parmeters (e.g. how many are the parameters to be predicted). The optimization problem to be solved seems to a hard one, since it involves both the GMM parameters and the network weights. The paper does not present convincing information about the viability of this task and how it depends on the initialization of the both the image clusters and the network parameters. In order to apply the method, the exact sequence of transformations should be specified. What happens if one or more transformations are redundant for a specific dataset?


Review 2

Summary and Contributions: ***Post rebuttal update*** I have read the author's rebuttal and thank the author for answering my questions. I am in favour of the paper's acceptance. This paper introduces a new method for clustering directly in image space. Existing methods build features on which to perform clustering in features space or use explicit image transformations to align the images before clustering in a joint optimisation manner. This paper also learns the transformations while clustering with a single loss and a joint optimisation algorithm, for both K-means and Gaussian Mixture Model (GMM). However, the authors propose to predict the transformations of each data point instead of optimising them, with use of a neural network. It thus builds on Spatial Transformers Networks and integrates the method in the clustering problem. Experiments are performed on standard benchmarks and more challenging real images (web images) to validate the relevance of their method.

Strengths: Strengths: * This work is not of theoretical contribution yet all claims are supported by strong empirical evaluation (ablation study is performed and extended comparison with existing methods is provided) and proof. * The method is novel albeit its comparison with relevant methods are missing (see below). * The significance of the paper is good, improving on minLoss over the different datasets, and interpretability. * This paper is of high relevance to the NeurIPS community as it is simple to implement, leads to interpretable results, and is shown to work on real web images.

Weaknesses: Weaknesses: * The paper experimental evaluation lacks an analysis (unless I missed it) of the effect of the number of clusters K, which is an important parameter of the model especially on real images when the number of cluster is unknown. This is an important point when deciding on the applicability of the method. * The related work section makes no comparison with the literature of equivariant models, that are, models that learn to encode the natural transformations of the data (e.g. rotations, translations), either with or without prior knowledge. See for example: https://arxiv.org/pdf/1901.11399.pdf (and references therein) https://arxiv.org/abs/1411.5908 https://arxiv.org/pdf/2002.06991.pdf

Correctness: Yes

Clarity: The paper is very clearly written and easy to follow.

Relation to Prior Work: Yes, but the paper lack a comparison with the literature of equivariant models (see point in weaknesses).

Reproducibility: Yes

Additional Feedback: * How do their method compares to transforming the samples x_i instead of the prototypes (apart from the fact that it would require to transform each data sample instead of the prototypes for the entire batch). * The authors claim that their method provide state-of-the-art results with a large margin on USPS and F-MNIST, in what sense ? MinLoss ? It seems that DEPICT and DCGAN are giving best avg results. * Incoherent notation: sometimes prototypes are written c_k and other times m_k * I would encourage the authors to explore the impact of the value of K


Review 3

Summary and Contributions: This paper presents a novel approach for transformation-invariant clustering, called Deep Transformation-Invariant (DTI). The main idea is to jointly learn image transformations (to align images) and to cluster them (previous work learn to cluster with explicit transformation). The main novelty of this work is to learn the transformation from the pixels while learning to cluster. The deep image transformation module is designed to learn image alignments . The module can model three types of transformation: spatial transforms (as in [34, 38]), color transform, and morphological ones (dilation, erosion). Experiments are conducted on standard benchmarks for image clustering, as well as web image benchmarks with strong results. Written presentation is clear and easy to understand.

Strengths: - The proposed method can simultaneously learn to align images and cluster which is new and interesting. - The design of the transformation module is interesting and include some new aspect of color & morphological transforms. - Experiments are strong compared with current methods.

Weaknesses: - The transformations are specifically applied to image data while the DTI framework can be more generic.

Correctness: I believe that most of the claims made in this paper are correct. The claim, in the final sentence in conclusion is not quite correct, since the transformations are particularly image-based transformations (spatial, color, morphological). In order to apply DTI to other type of data, one need to design data, domain-specific transformation module, while the DTI framework (objective, optimization, etc) can be the same.

Clarity: The paper is well written and easy to understand.

Relation to Prior Work: The paper covers enough relevant prior work.

Reproducibility: Yes

Additional Feedback: === post rebuttal comment === Rebuttal addressed well my comment, I keep my rating unchanged.


Review 4

Summary and Contributions: The paper proposes a novel approach towards deep image clustering which, unlike previous approaches does not aim at learning suitable latent space representations but at learning to predict image transformations in order to cluster in the image space. The proposed approach is a deep transformation-invariant clustering approach that jointly learns to cluster and align images. The transformations such as spatial alignment, color modifications or morphological transformations are learnt in an image transformation module. The paper provides a comparison to SotA image clustering approaches on MNIST, Fashion MNIST and USPS and show on par performance or small improvements.

Strengths: The proposed approach is conceptually different from existing approaches. And performs well. The idea to use transformers in this context is novel and interesting. The paper is clearly written and well illustrated. The proposed approach is evaluated in the context of kmeans clustering and Gaussian mixture models and performs well in both cases. The proposed approach provides interpretable qualitative results.

Weaknesses: The results depend on the initialization (kmeans), yet, they are reported without standard deviation. Mean, median and standard deviation over several runs should be reported. The improvement over the SotA is small.

Correctness: The paper is technically correct and the empirical methodology for evaluation is correct. The paper compares to the relevant literature. Yet, mean, median and standard deviation over several runs should be reported in table 1.

Clarity: Relevant competing approaches are discussed and compared to.

Relation to Prior Work: The relation to prior work is sufficiently discussed.

Reproducibility: Yes

Additional Feedback:

[Author Response · NeurIPS 2020]

# Deep Transformation-Invariant Clustering

We thank the reviewers (Rs) for their positive feedback. If accepted, we will incorporate all feedback in the final version.

**Lack of details on transformation parameters (R1).** We believe this information is already in the paper and supplementary material. As stated L160-162, we follow STN [34] to model the spatial transformations, i.e. we respectively model affine, projective and TPS transformations with 6, 8 and 16 (a 4x4 grid of control points) parameters. We will make this more explicit. As explained L163-168, the color transformation is modeled by an affine transformation with 2, 6 or 12 parameters depending on the scenario. The morphological module has 50 parameters ($a$, a $7 \times 7$ image and a real $\alpha$) as discussed L182-187. The exact sequence of transformations used for each experiment is specified in supplementary material Table 1. To ensure complete reproducibility, we will release code, data and models.

**Initialization (R1, R4).** Our results indeed depend on initialization, we will make this more explicit. We provided an analysis over 100 runs for MNIST in the supplementary material and report standard deviations for the experiments of Table 3. As suggested, we will add median and standard deviations for all tables. Note that, as discussed in the supplementary material L19-24, our loss correlates with performances and provides a way to select a good initialization.

**Small improvement over SotA (R4).** Although we do not report large quantitative improvements, our approach is very different from SotA methods, which we think is highly valuable. It has other significant benefits: (i) it doesn't rely on any hyper-parameter; (ii) the objective loss is simple to formulate and doesn't require any additional ad hoc losses or regularization terms; (iii) results are visually interpretable. Additionally, our method led to strong qualitative results on internet photo collections which, to the best of our knowledge, has not been reported by other approaches.

**Effect of the number of clusters $K$ (R2).** Similar to many clustering methods, the selection of the number of clusters is indeed a challenge. A purely quantitative analysis could be applied to select $K$, e.g. in Figure 1 we plot the average loss for DTI K-means as a function of the number of clusters and it is clear an elbow method could be applied to select 10 clusters. We believe our method also has the advantage to provide interpretable prototypes which we used to approximately select the number of clusters on the internet photo collections. We did not find the qualitative results on this data to be very sensitive to this choice, we will add loss plots, qualitative examples and a discussion.

Figure 1: Loss w.r.t nb of clusters for MNIST-test (avg over 5 runs)

**Equivariant models (R2).** They are indeed relevant, we will add a discussion in related work: our DTI framework learns to predict transformations of the prototypes in an equivariant way. We did not find any specific equivariant clustering method we could compare to, but would be happy to include comparison to any specific work. Note that equivariant models typically aim at learning equivariant representations, whereas our method aims at being invariant to transformations but not at learning a representation.

**Applications to other types of data (R3).** We will clarify the last sentence of the conclusion. To demonstrate our framework's genericity, we designed a 3D affine module (12 parameters) and applied our framework to cluster the 3D point clouds from ModelNet10 [1] (Table 1, results over 5 runs). Such a simple module already provides a significant boost over standard K-means.

Table 1: Cluster acc. on 3D shapes

| Method | avg | median | max | std |
|---|---|---|---|---|
| **K-means** | 72.1 | 73.7 | 74.1 | 2.4 |
| **DTI K-means** | **83.2** | **83.4** | **85.1** | 1.3 |

**Other comments (R1, R2).**

- *Hard GMM optimization (R1).* Our M-step is indeed more difficult than in standard GMM. We mentioned the training details used to make it work in Section 4.2 and in the supplementary material (Section D). We believe our experiments demonstrate the viability of our optimization.

- *Redundant transformations (R1).* Because transformations are all initialized with identity functions, adding redundant transformations in our curriculum learning process doesn't change the results. We experimentally validated it on MNIST using DTI K-means with 3 affine modules and the loss is not impacted when adding the duplicated modules.

- *Sample transformation (R2).* As discussed in the supplementary material (L51-57), a trivial solution if learning sample transformation instead of prototype one is to learn "empty" prototypes and transform samples into "empty" images. For MNIST, we rapidly observed black prototypes and any sample was transformed into a black image.

- *MinLoss criterion (R2).* We experimentally found that our loss is highly informative leading to a criterion (*minLoss*) that enables us to automatically select a high performances run in a fully unsupervised way (Section B of supplementary material). This is not trivial and to the best of our knowledge, such criterion has not been emphasized by other approaches like DEPICT or DSCDAN. Therefore, we argue performances obtained with *minLoss* can be compared with average results from competing methods which don't provide such criterion. We will clarify the claim.

- *Notations (R2).* We intentionally changed prototypes notations for the general framework ($c_k$) and its application to K-means ($m_k$) and GMM ($\mu_k$ and $\Sigma_k$). We will clarify.

[1] Z. Wu, S. Song, A. Khosla, L. Zhang, X. Tang, and J. Xiao. 3D ShapeNets: A deep representation for volumetric shapes. In *CVPR*, 2015.


[Meta-Review · NeurIPS 2020]

This paper presents a novel clustering method that learn image transformation in order to cluster images. The resulting approach is drastically different from existing work and could impact the future of the field in a positive way. The paper is well written and the experiments are convincing.